# What Flips the Switch? Signals and Stress Regulating Extraintestinal Pathogenic *Escherichia coli* Type 1 Fimbriae (Pili)

**DOI:** 10.3390/microorganisms10010005

**Published:** 2021-12-21

**Authors:** Hicham Bessaiah, Carole Anamalé, Jacqueline Sung, Charles M. Dozois

**Affiliations:** 1Institut National de Recherche Scientifique (INRS)-Centre Armand-Frappier Santé Biotechnologie, Laval, QC H7V 1B7, Canada; hicham.bessaiah@inrs.ca (H.B.); carole.anamale@inrs.ca (C.A.); chia-ling.sung@mail.mcgill.ca (J.S.); 2Centre de Recherche en Infectiologie Porcine et Avicole (CRIPA), Saint-Hyacinthe, QC J2S 2M2, Canada; 3Department of Microbiology and Immunology, McGill University, Montreal, QC H3G 0B1, Canada

**Keywords:** *Escherichia coli*, stress response, type 1 fimbriae, adhesion

## Abstract

Pathogens are exposed to a multitude of harmful conditions imposed by the environment of the host. Bacterial responses against these stresses are pivotal for successful host colonization and pathogenesis. In the case of many *E. coli* strains, type 1 fimbriae (pili) are an important colonization factor that can contribute to diseases such as urinary tract infections and neonatal meningitis. Production of type 1 fimbriae in *E. coli* is dependent on an invertible promoter element, *fimS*, which serves as a phase variation switch determining whether or not a bacterial cell will produce type 1 fimbriae. In this review, we present aspects of signaling and stress involved in mediating regulation of type 1 fimbriae in extraintestinal *E. coli*; in particular, how certain regulatory mechanisms, some of which are linked to stress response, can influence production of fimbriae and influence bacterial colonization and infection. We suggest that regulation of type 1 fimbriae is potentially linked to environmental stress responses, providing a perspective for how environmental cues in the host and bacterial stress response during infection both play an important role in regulating extraintestinal pathogenic *E. coli* colonization and virulence.

## 1. Introduction

The survival of microorganisms is highly dependent on their ability to adapt to a frequently changing environment and to respond to a variety of environmental cues. When important environmental changes occur, bacteria need to rapidly respond and adjust through complex transcriptional or post-transcriptional regulatory mechanisms [1]. Regarding bacterial pathogens, this adaptation is an important feature for survival and proliferation during colonization in different host anatomical niches that may represent distinct microenvironments. *Escherichia coli* is one of the best known and most studied free-living organisms. Pathogenic *E. coli* is commonly grouped into two broad categories: *E. coli* causing intestinal-type infections (InPEC) and those causing extraintestinal-type infections (ExPEC) [2]. This group of *E. coli* strains are phylogenetically and epidemiologically distinct from strictly commensal strains and other pathovars that cause intestinal diseases [3]. The ExPEC group brings together several pathotypes. Uropathogenic *E. coli* (UPEC) causes urinary tract infection (UTI) inducing cystitis, pyelonephritis, bacteremia, and sepsis. *E. coli* causing neonatal meningitis (NMEC) is responsible for meningitis in newborns [4,5]. Avian pathogenic *E. coli* (APEC) are the cause of avian colibacillosis, which manifests itself in various pathologies including aerosacculitis, pericarditis, perihepatitis, peritonitis, and sepsis [6]. These pathovars are a common cause of systemic infections in animals including humans [7,8,9]. Although ExPEC strains often possess and share many virulence strategies, the population genetics of clones belonging to ExPEC are also quite diverse [10]. ExPEC have a variety of virulence-associated traits, although there is no common set of genes responsible for ExPEC virulence, and many factors including the host background and immune status can contribute to disease outcome. During the infection process, the genome-wide transcriptional response of ExPEC during colonization of the host has provided insight into genes and virulence factors that may contribute to adaptation, such as iron and heme import systems, toxins, adhesins, lipopolysaccharides, invasins, capsules, and antibiotic resistance genes. These genes are encoded on mobile genetic elements, such as bacteriophages, transposons, plasmids, or in specific regions called Pathogenicity Islands, which are frequently found in ExPEC [8,11]. The structure of the *E. coli* genome consists of a flexible gene pool (including virulence genes) and a conserved part, which is also called the “core genome”. This core genome has been preserved throughout its vertical evolution, with very limited intragenomic rearrangement, resulting in the conserved synteny that is apparent today [12,13].

For pathogenic bacteria, the ability to adhere to host tissues is the initial step of infection. Adhesins are key virulence factors as they mediate interactions with host cells or mucosa and promote bacterial colonization and infection. Adhesion is even more important during infection where microenvironmental stresses such as changes in pH, osmolarity, temperature, and mechanical forces are encountered. ExPEC are able to express a large variety of adhesins such as P fimbriae, curli fimbriae, or type 1 fimbriae, with different receptor specificities [14]. For example, UPEC can express more than 10 different types of fimbriae, particularly the UPEC strain CFT073 has at least 12 distinct fimbriae and several afimbrial adhesins [12,15]. This spectrum provides the bacterium with the capacity to bind to a range of different target molecules; however, fimbriae are important immunogenic factors and therefore, bacteria have no interest in expressing more than one at the same time. During colonization, their expression is subject to “regulatory crosstalk”, which allows bacteria to express the appropriate fimbriae at the right time [16]. As an example, Snyder and others have demonstrated that in a UTI, when type 1 fimbriae are overexpressed in vivo, P fimbriae expression is downregulated. In addition, in the UPEC strain CFT073 expressing neither type 1 fimbriae nor P fimbriae, FIC fimbriae expression was increased [17]. In some literature, fimbriae may also be referred to as pili. Both terms can be designated to adhesins [18]. However, in this review, the long filamentous non-flagellar structures allowing bacterial adhesion to cells are referred to as fimbriae, whereas the term “pili” is reserved for F or conjugative pili involved in bacterial mating [14,19].

### Role of Type 1 Fimbriae in Pathogenesis

One of the most important virulence factors of pathogenic *E. coli* is type 1 fimbriae. This fimbrial adhesin can mediate bacterial attachment to and invasion of host cells and is subject to regulation through phase variation by a variety of environmental signals. More specifically, bacterial attachment via type 1 fimbriae to host d-mannosylated proteins will trigger signal transduction and induce actin rearrangement in target cells, allowing the pathogen to invade. In UPEC, type 1 fimbriae bind to the mannose-enriched uroplakins found on urothelial cells of the bladder [12]. Once internalized, the bacteria can rapidly multiply to form biofilm-like intracellular bacterial communities (IBCs) where they can evade host immune defenses and antibiotic treatments. As they proliferate, bacteria can then disperse from IBCs to colonize and invade other cells. IBC formation mediated by type 1 fimbriae is especially important for UPEC pathogenesis as it promotes bacterial ascension from the urinary tract to the kidneys [20]. Although the role of type 1 fimbriae has mainly been studied in UPEC, the fimbrial adhesins have also been shown to contribute to NMEC pathogenesis through adherence and invasion of human brain microvascular endothelial cells (HBMEC) [21]. In APEC strains, type 1 fimbriae are associated with survival, fitness, and pathogenesis by allowing more colonization of the trachea and the lung [22,23].

The following sections will present the current state of knowledge for general and specific regulators of stress known in ExPEC and the impact of type 1 fimbriae regulation.

## 2. Type 1 Fimbriae

### 2.1. Type 1 Fimbriae Biogenesis

Fimbriae (pili) are long, proteinaceous organelles that extend from the surface of many bacteria and mediate diverse functions, including attachment, invasion, and biofilm formation. In Gram-negative bacteria, fimbriae are assembled via a range of different protein translocation systems, including the chaperone-usher (CU) pathway, the type IV secretion pathway, and the extracellular nucleation precipitation pathways [24].

Chaperone-usher fimbriae (CUF) are morphologically characterized as being relatively thick (~7 nm diameter), rod-like fibers with a length varying between 0.2 and 2 μm [25]. CU fimbriae are comprised of multiple copies (>1000) of the major fimbrial subunit and a tip adhesin that is linked by an adapter complex, which often consists of multiple minor subunit proteins [26]. Fimbrial subunits are shuttled through the inner membrane to the periplasm by the general secretory pathway, SecYEG translocon [27]. These subunits are then linked together via a zip-in zip-out mechanism coordinated by periplasmic chaperone proteins and a pore-forming usher protein, which acts as a scaffold for subunit assembly [28]. The chaperone facilitates several essential steps in the pathway; it mediates the folding of fimbrial subunit proteins, prevents their polymerization in the periplasm, and directs their passage to the usher. The usher in turn acts as an assembly platform and facilitates the assembly of the fimbrial structural organelle (structural component of a fimbria). Briefly, the *N*-terminal extension on an incoming fimbrial subunit displaces the beta-strand of the chaperone protein bound to the previously assembled subunit. Through this mechanism of strand exchange, fimbrial subunits are rapidly polymerized to form fimbriae [29].

Fimbrial adhesins, which are often located at the tip of the organelle, typically recognize specific receptor targets in a lock-and-key fashion, thus enabling the bacterium to target a specific surface and display tissue tropism.

### 2.2. Genetic Organization of Fimbrial Gene Clusters and Transcriptional Regulation

Type 1 fimbriae are among the most common adhesins in *E. coli* and are encoded by the *fim* gene cluster [30]. Nine genes encode the structural components and specific transport systems (*fimAICDFGH*), and the regulatory genes (*fimB* and *fimE*) [30,31,32]. FimA is the major structural subunit, which forms the majority of the extracellular filament. FimC and FimD are the chaperone and the usher, respectively, that facilitate the transport of the subunits to the bacterial surface. FimH, the adhesin tip, is integrated into the organelle structure with the help of adaptors, FimF and FimG. Although *fimI* is part of the operon, its function/role remains unknown; however, it is required for biogenesis of fimbriae [33,34,35,36,37].

The expression of type 1 fimbriae is governed by the orientation of a 314 bp invertible element (the *fim* switch), located immediately upstream of the major subunit gene and flanked by two 9 bp inverted repeats (5′ TTGGGGCCA) [34]. The expression of type 1 fimbriae is phase-variable, meaning that the promoter located within an invertible element (IE) *fimS* can switch between two different orientations. The phase-ON orientation (fimbriated phenotype) of the IE allows transcription of *fimA* and other accessory genes, resulting in the expression of type 1 fimbriae. When *fimS* is in the opposite orientation, no type 1 fimbrial transcription occurs, and bacteria are phase-OFF (type 1 fimbriae-negative). The inversion of the element is mediated by two site-specific recombinases, FimE, which primarily promotes switching from phase-ON to phase-OFF, and FimB, which can mediate switching in either direction [31,38].

One of the earliest phenotypic characterizations of type 1 fimbriae was their ability to confer d-mannose-sensitive hemagglutination of guinea pig erythrocytes [12,39]. Further characterization of the type 1 tip adhesin, FimH, demonstrated that type 1 fimbriae recognize mannose, which is found on the surface of many types of host cells.

## 3. Regulators of Stress Responses and Type 1 Fimbriae in ExPEC

The stress response can be defined as the change in gene expression of bacteria for an optimal environmental adaptation. These changes can be controlled by a specific sigma factor (e.g., master regulator, heat shock response) or another transcriptional regulator (e.g., SoxR/S or OxyR), a two-component system, the nutritional starvation response (the stringent response), or small RNAs [40]. These regulators can mediate changes in bacterial gene expression to adapt to stress. Some affected genes are implicated in virulence, such as type 1 fimbriae. As type 1 fimbriae play a key role in mediating *E. coli* host colonization and virulence, it is important to understand the regulation of these fimbriae in relation to stress responses. Indeed, numerous regulators (Figure 1) and growth conditions (Figure 2) have been identified that can affect the production of type 1 fimbriae. Below we present regulators that can play important roles in global stress regulation but have also been shown to affect expression of type 1 fimbriae (Table 1 and Table 2).

### 3.1. General Stress

**RpoS (Sigma S, σS):** The sigma factor σS (also called RpoS or σ38) is one of the most studied global regulators of the general stress response (Table 1). The σS-mediated general stress response can be defined as the induction of various mechanisms to prepare bacterial cells for increased stress conditions, including toxicity and stress due to population density during culture, acidity, oxidative, and osmotic stresses, often described as the stationary-phase stress response [54,55]. Bacterial gene transcription requires RNA polymerase (RNAP) which binds to a dissociating σ factor (σ) to initiate the transcription process. Alternative σ factors also exist and are often linked to specialized regulators that are activated during growth transitions, morphological changes, or under specific stress conditions (nitrogen metabolism, heat shock, iron-limitation) [56]. In *E. coli*, RpoS is considered the master regulator in the general stress response [41]. The *rpoS* gene, which encodes σS, regulates, directly or indirectly, at least 500 genes (more than 10% of the *E. coli* genome) implicated in survival during the stationary phase and resistance to stress. Although RpoS can be considered as a master general regulator, regulation of RpoS is also influenced by other regulators [57,58]. For example, Girard and others have shown that the synthesis of σS is positively influenced by the transcription factor, DskA, and its cofactor, the ppGpp alarmone (Figure 2) [59].

Although one of the main adaptations to environmental conditions involving RpoS is the transition from exponential growth in a nutrient-rich environment to the stationary phase, which can lead to accompanying nutrient-deficient conditions, RpoS is also critical for regulation of oxidative stress and expression of catalase, e.g., *katG* and *katE* in *E. coli* (Figure 2) [60]. For instance, during urinary tract infection, σS is required for UPEC strain CFT073 to withstand phagocyte-mediated oxidative stress [61] and acid stress via expression of *gadX* (Figure 2) [62]. RpoS also plays a role in UPEC via its involvement in the biosynthesis of the signal molecule, c-di-GMP, promoting the production of adherence and cell-aggregating factors including type 1 fimbriae while reducing cellular motility by repressing the production of flagella [63,64]. RpoS is directly involved in regulating the expression of virulence factors in other ExPEC such as strains responsible for neonatal meningitis. In meningitis strain *E. coli* K1, RpoS contributes to invasion of brain microvascular endothelial cells [65].

Regarding type 1 fimbriae, RpoS is activated as bacteria enter the stationary phase and represses *fimB* transcription by potentially affecting its promoter (Table 2). In *E. coli* K-12, by repressing *fimB* transcription (Figure 1), all bacterial cells will gradually turn off type 1 fimbrial expression. Specifically, it was demonstrated that RpoS mutants did not have type 1 fimbrial repression during the stationary phase. Further, *rpoS* mutants had an accelerated phase variation frequency; and this is most likely due to the increased expression of *fimB* [66]. However, other studies have demonstrated that *fimA* expression stays relatively high during the stationary phase, a contradicting result to the ExPEC fimbrial repression by RpoS [67,68]. Similarly to RpoS, H-NS is also highly growth-phase dependent and even though both belong to their own independent pathways, H-NS demonstrates some regulatory function over RpoS. Zhou and others have demonstrated that H-NS indirectly activates SprE, a response regulator that is part of the RpoS degradation pathway. Further, H-NS mutants have an increase in RpoS stability. In other words, H-NS reduces RpoS levels which may have an indirect effect on type 1 fimbriae regulation [69].

**LrhA:** LrhA is a transcriptional regulator of the LysR family involved in flagellar motility and fimbrial genes. Regarding type 1 fimbriae, LrhA indirectly affects *fimS* phase variation by promoting *fimE* transcription (Table 2). By analyzing quantitative RT-PCR results, Blumer and others have demonstrated that levels of *fimE* transcripts decrease in LrhA mutants, leading to increased type 1 fimbrial expression. Further investigation showed that even though LrhA can bind to promoter regions of *fimA*, *fimB*, and *fimE*, it has the highest affinity for the *fimE* promoter and only directly affects *fimE* transcription. In other words, these results suggest that LrhA represses type 1 fimbrial expression by promoting *fimE* transcription [73]. Interestingly, LrhA also functions as a regulator of RpoS. More specifically, it promotes RpoS degradation by affecting the activity of the protease, ClpXP [87]. Therefore, LrhA also indirectly links environmental stress and the expression of type 1 fimbriae through modulation of the RpoS network.

**H-NS:** The histone-like nucleoid structuring (H-NS) protein is a major component of the enterobacterial chromosome with roles in many biological processes. H-NS is known to modulate the expression of at least 200 genes in *E. coli* [88,89,90]. Since H-NS mostly affects transcription negatively, this protein is considered a transcriptional repressor (Table 1) [90,91]. In addition to a repressor role at the transcriptional level, studies have shown that H-NS can exert a positive and negative effect at the post-transcriptional level and influences bacterial virulence [72]. H-NS mainly negatively regulates genes whose expression is influenced by environmental factors, such as temperature (including cold and heat-shock), oxygen availability, osmolarity, growth phase, or pH [90,92,93]. For example, it has been shown that *hns* mutants have an improved chance of survival under extremely acidic conditions and increased transcription of the global regulator, RpoS [69,94,95].

In pathogenic *E. coli*, the relationship between temperature and H-NS regulation has been studied in the control of specific genes associated with virulence traits including motility, toxin production, pathogenicity island-associated genes, or fimbriae (pili) [96,97]. For example, H-NS positively affects the synthesis of flagella, by interacting directly with the *flhDC* regulatory region [42]. At low temperatures, the H-NS protein was shown to repress transcription of fimbrial operons, including the *fim* and the *pap* operon [98,99].

Regarding the *fim* operon, H-NS may have a direct effect on promoter switching. Studies have shown that H-NS can bind to DNA segments adjacent to *fimS* [68]. However, the indirect pathway of which H-NS regulates phase variation is better understood. Generally, H-NS represses *fimS* switching by repressing *fimB* and *fimE* expression (Figure 3). The protein potentially binds to *fimB* and *fimE* promoters, which can either block RNA polymerase from binding or prevent the latter from properly functioning (Figure 1 and Figure 3) [97]. Different studies have shown with H-NS mutants that H-NS regulates genes required for stress resistance and that different environmental conditions can affect H-NS expression [100]. However, it remains unclear how these conditions affecting H-NS influence type 1 fimbrial expression. Interestingly, Olsen and others demonstrated with strains that constitutively express H-NS, that when the temperature was increased from 30 to 37 °C, *fimE* promoter activity reduced (less OFF-phase cells) whereas *fimB* promoter activity increased (more ON-phase cells) [72]. The difference in *fim* expression may be due to the higher affinity of H-NS to the *fimB* promoter (Table 2). In other words, at a lower temperature, H-NS expression is induced by the cold-shock protein, CspA, indicating a higher expression of H-NS and higher repression of *fimB*, resulting in fewer ON cells. These results suggest that H-NS favors type 1 fimbrial expression at mammalian body temperature, indicating that H-NS is able to sense the optimal environment and respond to these signals. Further, H-NS affects the Lrp pathway by repressing Lrp expression. Not only were *hns* null mutants found to have an increased type 1 fimbrial expression, but they also had an increased level of Lrp [101]. Therefore, induction of H-NS expression could lead to an antagonized Lrp expression, which decreases *fim* phase variation frequency.

**Lrp:** The leucine-responsive regulatory protein (Lrp) is an abundant regulatory protein that usually links bacterial metabolism to environmental signals, especially in nutrient-deficient conditions [102,103]. More specifically, growth in nutrient-poor medium leads to higher expression of *lrp*, which can act as a global regulator to activate transcription of genes that are part of the Lrp regulon. (Table 1). Depending on the target, effects can be potentiated, repressed, or unaffected by the level of exogenous leucine [43,104] (Figure 2). Although activation or repression by Lrp can be potentiated by leucine, Lrp-mediated regulation can also occur independently of the presence of leucine [105].

In *E. coli*, Lrp regulates ≈ 400 genes of which over 200 genes are involved in direct interactions with target DNA sequences. Indeed, Lrp has been shown to regulate ≈10% of all ORFs in *E. coli* [103,106,107]. In order to survive in various environments while avoiding energy loss, *E. coli* must respond to environmental changes, such as nutrient levels, by regulating the transport of amino acids. Lrp does so by regulating the expression of many amino acid exporters. Indeed, due to limited nutritional resources, bacteria must stop replication and enter the stationary phase [102,108,109]. Further, Lrp also regulates the production of many types of fimbriae, including *pap*-encoded P fimbriae and *fim*-encoded type 1 fimbriae [43]. Similarly to IHF, Lrp is another site-specific DNA-binding protein that promotes both *fimB*- and *fimE*-mediated phase switching (Figure 3). Mutants that have defective Lrp or modified binding sites demonstrated lower phase-switching frequencies. In type 1 fimbriae, leucine potentiates Lrp activity, resulting in more phase variation (Table 2) [102]. Upregulation of Lrp occurs when bacteria are found in a nutrient-deficient environment, which is dependent on the bacterial growth phase. Its expression tends to increase as bacteria enter the stationary phase, where nutrient levels gradually diminish [110]. Therefore, nutritional stress will result in upregulation of Lrp, which will then stimulate phase variation. Interestingly, the presence of other amino acids such as alanine, isoleucine, and valine were found to potentiate Lrp activity in *fimB*- and *fimE*-mediated phase switching of type 1 fimbriae production [71].

**IHF:** The integration host factor (IHF) is a heterodimeric site-specific DNA-binding protein, composed of an IHFα subunit and an IHFβ subunit [70]. IHF responds to environmental changes, more specifically, when bacteria enter the stationary phase or find themselves in a nutrient-deficient environment [111]. Regarding the *fim* switch (Table 2), IHF induces a sharp bend that facilitates the formation of a synapse between the inverted repeats, creating a recombination-promoting structure. This structure allows a more efficient phase switching (Figure 3). Additionally, IHF was found to bind to DNA segments adjacent to and within the *fim* switch. When these binding sites were mutated, leading to a lower affinity of IHF, the phase-switching frequency decreased dramatically as well. These results indicate that IHF acts directly on phase variation and when the bacteria are in starving conditions, an increased expression of IHF will promote *fimS* switching, in whichever orientation that favors bacterial survival (the exact orientation remains unclear) [70].

**ppGpp**: The small nucleotides (p)ppGpp, also called alarmones, are a term related to two nucleotides: ppGpp and (p)ppGpp. This guanosine penta- and tetraphosphate is the effector molecule of the well-known stringent response, which was first described in 1996 [112] and represents a general bacterial stress response [113,114]. During nutrient starvation (Table 1), the accumulation of ppGpp allows bacterial cells to readjust their metabolism and physiology to slow down growth. The accumulation of ppGpp plays a role in response to various stressful signals including osmotic shock, temperature variation [115], oxidative conditions, or pH downshift [116]. The molecule also takes part in biological processes, such as biofilm formation [44], antibiotic resistance [115], production of surface organelles such as flagella [45], and as previously mentioned, the general stress response [117].

In *E. coli*, when the amino acid levels become limiting for growth, ppGpp binds to RNA polymerase (RNAP). This interaction causes downregulation of stable RNAs (rRNA and tRNA), restricting protein synthesis [118,119,120,121,122]. Additionally, studies have shown that the accumulated alarmone can influence the transcription of ≈500 genes [123,124]. During the stringent response, type 1 fimbriae-encoding gene expression is down-regulated. Indeed, ppGpp disrupts *fimB* transcription by affecting activity at one of the three *fimB* promoters (Figure 1), leading to a decrease in type 1 expression (Table 2). Aberg and others have demonstrated that ppGpp mutants showed almost no yeast agglutination, indicating an absence of type 1 fimbriae production. In addition, using a *fimA*-*lacZ* fusion, it was shown that *fim* transcription levels were also dramatically reduced [67].

**c-AMP:** The 3′, 5′-cyclic adenosine monophosphate (cAMP) is a ubiquitous molecule present in both prokaryotes and eukaryotes [125,126,127]. The cAMP receptor protein (CRP) is the target for cAMP signaling and is also capable of regulating genes involved in the catabolism of secondary carbon sources [68,128]. cAMP was initially described for its role in mediating the ‘glucose response’, or carbon catabolite repression (CCR) [129,130]. In bacteria, this term is related to a regulatory mechanism that allows a specific and important utilization of carbohydrates. In addition to its known role in metabolic regulation, cAMP has an extended function in global gene regulation. A misregulation of cAMP signaling for carbon availability can affect expression of key virulence influencing host colonization [131].

In *E. coli*, the cAMP–CRP complex serves as a global transcriptional regulator of the expression of ≈200 genes [132]. CRP also can play a role in acid stress response, when during exponential growth in rich medium, it can repress the RpoS-dependent *gad* gene transcription and contribute to bacterial survival in acidic environments [46,62,133]. Further, intracellular cAMP levels affect the transcription of OxyR, a regulator contributing to the response against oxidative stress (Figure 2) [134,135] or intracellular cAMP levels can also be modulated by external osmolarity [136,137]. Interestingly, the effects of cAMP signaling are often amplified by cAMP-mediated co-regulation of other global regulators such as RpoS [138]. Generally, cAMP–CRP mutants are attenuated due to their hypersensitivity to reactive nitrogen species and inability to utilize a number of carbon sources such as lactose or amino acids. However, these mutants are highly resistant to hydrogen peroxide and acid stress and in a way, have a greater survival advantage. In fact, cAMP–CRP mutants had an increased expression of RpoS, which led to an increase in the production of catalase to respond to environmental stress. In other words, cAMP participates in the regulatory pathway of RpoS by repressing its transcription and therefore also indirectly regulates type 1 fimbrial expression [139].

With regard to the direct effect on production of type 1 fimbriae, CRP–cAMP represses *fimB*-mediated recombination (Table 2). Mutants that were *fimB* proficient demonstrated an increased population of Fim-ON cells in the absence of CRP–cAMP; however, in mutants that were *fimE* proficient, no significant differences were observed [68]. In both regulatory pathways, the high concentration of cAMP, indicating nutrient deprivation, led to the repression of type 1 fimbriae.

Furthermore, Muller and others have demonstrated that CRP–cAMP modulates *gyrA* (DNA gyrase). The study showed that CRP mutants had lower *gyrA* expression and gyrase activity, which may explain the repression of *fimB*-mediated recombination. In fact, CRP–cAMP mutants had similar levels of *fim*-ON cells as bacteria treated with novobiocin, a DNA gyrase inhibitor. There are also other factors, such as Lrp (see above), that aid with the recombination of *fimS*. In CRP mutants, Lrp levels were found to be increased. Potentially, due to the decreased expression and activity of DNA gyrase in *crp* mutants, Lrp is up-regulated to compensate for the loss [68]. In summary, CRP–cAMP is able to regulate type 1 fimbriae expression by indirectly regulating the RpoS and Lrp pathways as well as by directly modulating DNA gyrase activity that potentially affects *fimB*-mediated recombination.

**Regulation by sRNAs**: Bacterial small regulatory RNAs (sRNAs) are commonly 50 to 500 nucleotides long and are involved in various stress responses to environmental changes including those to overcome membrane damage, pH variations, and oxidative stress. In fact, sRNAs mediate the regulation of regulatory proteins by affecting their transcription or their activity. Mechanisms employed by bacterial sRNAs can be divided into two categories: the cis-encoded sRNAs, which are coded by the complementary DNA strand of their mRNA targets, and the trans-encoded sRNAs which are located remotely from their mRNA targets and often exhibit only partial complementarity to them. The well conserved Hfq protein chaperone is required as a cofactor for these RNA–RNA interactions to facilitate sRNA stabilization [140,141,142]. Regulation by Hfq can lead to an upregulated expression by remodeling inhibitory RNA structures or blocking access of negative regulators such as RNases or Rho but can also downregulate expression by recruiting Rnases [143]. Therefore, a given sRNA can have various mRNA targets and can carry both positive and negative regulatory roles. Likewise, an individual mRNA can be a target of multiple sRNAs. This is the case of RpoS whose expression is regulated by at least three different sRNAs: OxyS, DsrA, and RprA.

The genome of *E. coli* comprises at least 90 sRNAs of which a large number is implicated in bacterial virulence [144]. Several sRNAs are involved in membrane stress [49]. sRNAs also play a role in nutrient stress response, such as RyhB, which is an sRNA expressed under iron-starvation conditions and is regulated by the Fur protein [145]. In UPEC, RhyB facilitates the synthesis of many iron-scavenging siderophores including enterobactin, salmochelin, and aerobactin, further suggesting its role as a virulence mediator of UTI in animal models [50]. Regarding pH stress, the Hfq-dependent GadY sRNA is mostly involved in acid stress resistance in *E. coli* (Table 1) [51]. Recent work on the sRNA RyfA from UPEC strains revealed its roles in resistance to oxidative and osmotic stresses and survival in human primary macrophages. RyfA will be further discussed in later sections [52].

**RpoH (σ32) and Heat shock (HS):** The heat shock response (HSR) is defined as a cellular response to sustain protein homeostasis and promote heat resistance in eukaryotic and prokaryotic cells [146,147,148]. Thus, elevated temperature or other environmental cues that disturb protein homeostasis induce the accumulation of misfolded proteins and lead to the transcription of genes encoding heat shock proteins (HSPs) such as DnaK/DnaJ and GroEL/GroES chaperones (Figure 2) [149,150,151]. In *E. coli*, the HSPs are regulated by *rpoH* which encodes the alternative sigma factor, σ32 (RpoH) [152]. In *E. coli*, HSR is induced by a shift from 30 °C to 42 °C through transcription of heat shock genes (*hsp*) which are regulated by this temperature upshift (Table 1). RpoH initiates the transcription of ≈90 genes [53]. DnaK plays a role in the pathogenicity of multidrug-resistant bacteria such as in *E. coli* K-12 W3110. A *dnaK* mutant showed a strong susceptibility to fluoroquinolones [153]. However, the regulation of heat shock genes is complex and still needs to be clarified.

### 3.2. Envelope Stress

**CpxRA two-component system:** In Gram-negative bacteria, the bacterial envelope is an important interface between the bacterial cell and the often stressful extracellular environment [154]. This complex envelope protects bacteria from harsh conditions and must be able to endure stresses such as acidic or basic pH, antimicrobial cationic peptides, bile, perturbations caused by misfolded proteins, and alterations in phospholipids and lipopolysaccharides [155]. The bacterial envelope integrity is mostly dependent on Envelope Stress Responses (ESRs) which can sense the presence of extracellular stress and the disruption of homeostasis in the periplasm. The ESRs are regulated by two-component systems (TCS) which include the Bae, Rcs, and Cpx systems, or by RNA polymerase-associating alternative sigma factors [156,157].

In *E. coli*, the best characterized ESRs are regulated through the alternative sigma factor σE (response to stress in outer membrane/periplasm) and the TCS system CpxRA (response to stress in the inner membrane) (Table 1) [158]. Interestingly, the CpxRA pathway mediates the transcription of genes involved in cell adhesion, biofilm formation, and antibiotic resistance (Figure 2) [159,160]. Further, the pathway mediates the regulation of the expression of genes involved in surface structures associated with bacterial virulence such as type three secretion systems (TTSS) or adhesive organelles and fimbriae [48].

In APEC strain MT78, Matter and others have demonstrated with *cpxA* mutants that the system negatively affects type 1 fimbrial expression. However, the double *cpxRA* mutants showed type 1 fimbriae expression similar to the wild-type levels. This result indicates that CpxR-P is directly bound to the *fimA* promoter region leading to a phase-OFF orientation (Table 2) [74]. Conversely, in zebrafish and murine models and *cpxRA* mutants from UPEC strains UTI89 and CFT073, the deletion of the *cpxRA* operon led to decreased colonization of the murine bladder and reduced virulence [161]. At this point, there is a possible link between the Cpx envelope stress response and type 1 fimbriae expression, but the *fim* operon has not been reported to be directly regulated by CpxR in these UPEC strains.

**BarA/UvrY:** In *E. coli*, the TCS BarA/UvrY controls carbon metabolism, flagella, and biofilm formation by regulating the activity of CsrA. In the APEC strain χ7122, a *barA* or *uvrY* mutant demonstrated reduced *fimA* expression (two-fold compared to the wild-type) [75]. Further, BarA/UvrY TCS is also known to regulate RpoS. The study done by Herren and others, shows evidence for decreased transcription of *rpoS* in *barA* and *uvrY* mutants. This repression leads to a down-regulation of the *pst* operon, ref [162] whose inactivation in UPEC strains has also been reported to reduce transcription and production of type 1 fimbriae [81]. Therefore, it appears that BarA/UvrY is a global regulator in APEC strains, which can indirectly affect type 1 fimbrial expression by regulating the *rpoS* gene (Figure 1); however, the exact link between BarA/UvrY and regulation of type 1 fimbriae remains unclear.

**OmpA:** Teng and colleagues highlighted the potential role of the outer membrane protein A (OmpA) in the NMEC strain *E. coli* K1. Therefore, due to its importance to maintain the integrity of the bacterial outer membrane structure, loss of OmpA could induce an envelope stress response. Studying yeast agglutination and expression of *fim* gene, it has been shown that, in vitro, *ompA* deletion in *E. coli* K1 decreased the expression and production of type 1 fimbriae. Further, in in vivo experiments, *ompA* mutants exhibit reduced ability to bind and invade HBMEC. However, this decrease may not be completely due to diminished type 1 fimbrial expression. Although the role of type 1 fimbriae in the pathogenesis of meningitis *E. coli* remains to be clarified, this study suggests that OmpA and type 1 fimbriae potentially contribute to *E. coli* K1-associated meningitis [163].

### 3.3. Osmotic and Oxidative Stress

Oxidative stress can be described as an excess of cellular prooxidants. Oxygen molecules such as superoxide (O_2_•^−^), hydrogen peroxide (H_2_O_2_), hydroxyl radical (•OH), nitric oxide (NO), and other oxygen-derivative intermediates that can modify organic molecules are referred to as reactive oxygen species (ROS). Oxidative stress is caused by bacterial respiratory activity or by toxic molecules released by host cells. Host phagocytes can generate RO, exerting antimicrobial activities against a broad range of pathogens [155,164,165].

A number of studies have demonstrated a link between osmotic stress and type 1 fimbriae, some of which have been contradictory. The discrepancies observed may be due to regulatory differences in strains used in experiments as well as the variability in composition of urine samples. For example, Schwan et al. demonstrated in UPEC strain NU149 that osmotic stress caused by NaCl, and acidic conditions induced a decrease in type 1 fimbriae expression [166]. By contrast, Snyder et al. demonstrated that UPEC strain CFT073 had an increased level of type 1 fimbriae expression during UTI [167]. Further, Withman et al. also showed that type 1 fimbriae expression by strain in CFT073 increased under osmotic stress due to an increase in urea whereas increased NaCl concentration had no effect on levels of type 1 fimbriae [168]. Conversely, a report by Greene and colleagues showed that UPEC strain UTI89 had decreased expression of type 1 fimbriae in human urine [169].

**SoxS/R and OxyR:** The oxidative stress response is mainly mediated at the transcriptional level through two major regulatory systems, OxyR and SoxRS (Table 1) [170]. These two transcriptional regulators belong to different families. OxyR, a LysR family transcriptional factor, regulates genes that play a role in the removal of hydrogen peroxide, whereas SoxRS regulates genes that target superoxides and nitric oxides [171,172,173,174]. The OxyR protein is composed of a regulatory domain, which senses H_2_O_2_ concentrations, and a DNA-binding domain that can directly control gene expression. The SoxRS system includes a redox sensor/regulator, SoxR, and a second regulator, SoxS which subsequently regulated ≈100 genes that help bacteria withstand O_2_^−^ products [170].

*E. coli* uses the SoxRS and OxyR systems to resist stress caused by high levels of ROS (Figure 2). Members of the SoxRS regulon control the expression of mainly the superoxide dismutases *sodA* and *sodB* [175]. Additionally, the OxyR regulon induces the transcription of genes that increase resistance to hydrogen peroxide, including *katG* and *katE* (catalase), *ahpCF* (alkylhydroperoxide reductase), and *dps* (iron sequestration) [176,177]. SoxRS regulon also regulates genes that modify lipopolysaccharide (LPS) in the cell envelope of *E. coli*, which affects its resistance against antibiotics [178]. Likewise, in UPEC, inactivation of *oxyRS* leads to an increased susceptibility to H_2_O_2,_ ref [179]. Deletion of *oxyRS* in the wild-type UPEC strain *E. coli* Ec1a and CFT073 resulted in decreased resistance to H_2_O_2_ and decreased virulence in a mouse model of ascending UTI [47]. Although SoxRS and OxyR have not been reported to influence type 1 expression, their pathways share intermediary factors that are implicated in type 1 fimbriae regulation, suggesting they may exert an indirect effect on type 1 fimbriae as well.

**TreA:** TreA encodes a periplasmic trehalase which hydrolyzes trehalose, a key osmoprotectant molecule in bacteria. In line with the link of osmotic stress to the expression of type 1 fimbriae, Pavanelo et al. reported that deletion of *treA* from ExPEC strain MT78 leads to an increase in osmotic resistance to urea as well as a decrease in expression of type 1 fimbriae. Loss of TreA also led to reduced colonization of the uroepithelium in a murine UTI model [76].

**YeaR:** Recent studies have shown that *yeaR*, encoding a protein of unknown function, could have a direct influence on the formation of intracellular bacterial communities (IBCs) by UPEC strain UTI89. Interestingly, *yeaR*, an uncharacterized gene, is overexpressed in IBCs. Conover et al. have shown that a *yeaR* mutant of UTI89 also has decreased *fim* expression (Figure 1). Type 1 regulation is linked to changes in oxidative stress via YeaR, which can potentially induce a yet-identified stress regulator that directly influences *fimS* switching [77].

**Other cytoplasmic and periplasmic proteins:** IbeA (invasion of brain endothelium) is encoded by the GimA genomic island present in some *E. coli* strains. The role of GimA in the pathogenesis of newborn meningitis and ExPEC is well characterized. Recent studies showed that in an APEC strain BEN2908, a *ibeA* mutation caused a decrease in the expression of the *fimB* and *fimE* recombinases and reduced expression of type 1 fimbriae [78]. Further, IbeA is involved in resistance to oxidative stress of pathogenic *E. coli* strains by increasing H_2_O_2_ resistance [180]. It is likely that IbeA plays a role in modulating type 1 fimbriae expression through oxidative stress; however, the mechanism remains unclear. Similarly, we recently demonstrated that in UPEC strain CFT073, YqhG, a predicted periplasmic protein contributes to virulence in the urinary tract inducing a decrease in the production of type 1 fimbriae. In addition, this protein is involved in resistance to oxidative stress (Figure 1) [79].

**Small RNA RyfA:** Another aspect of oxidative resistance has been demonstrated with the small RNA *ryfA*. As previously discussed, *ryfA* is involved in the regulation of resistance to oxidative and osmotic stresses and its deletion reduced UPEC survival in human macrophages. RNA-seq analysis revealed that genes involved in survival or virulence were downregulated in a *ryfA* mutant in addition to multiple operons encoding fimbriae. In the mouse UTI model, inactivation of *ryfA* in UPEC strain CFT073 showed a decrease in urinary tract colonization. The *ryfA* mutant also had reduced production of type 1 (Figure 1). Taken together, the results suggest that *ryfA* may play a key regulatory role in UPEC adaptation to oxidative and osmotic stress. The specific contribution that this small RNA plays in the regulation of type 1 fimbriae is currently unknown (Table 2) but studies are in progress to determine the pathways and the mechanisms of how *ryfA* acts on *fim* switching [52].

### 3.4. Nitrosative Stress

Prior studies have identified FimX as a DNA invertase (Fim-like family member associated with ExPEC) that regulates *fim* expression in the urinary tract by mediating the phase OFF to ON transition [181]. FimX is also an epigenetic regulator of a LuxR-like response regulator, HyxR. More specifically, FimX epigenetically regulates the expression of *hyxR* via bidirectional phase reversal of its promoter region at sites different from the *fim* promoter. In addition, the expression of HyxR leads to a suppressed tolerance and survival in the presence of reactive nitrogen intermediates (RNI). The ability of UPEC UTI89 to survive RNI-mediated stresses in macrophages depends on the proper regulation of HyxR, which acts as a negative regulator of RNI response pathways. In the study done by Bateman et al., it was observed that HyxR could repress the expression of a bacterial nitric oxide detoxification enzyme and therefore resist nitrosative stress [80]. Furthermore, FimX produces unidirectional phase inversion of the *fimS* promoter, preventing the expression of type 1 fimbriae. These observations suggest that FimX may be important for mediating the inversion of both the *fimS* and *hyxR* promoters. In summary, FimX may coordinately regulate crosstalk between nitrosative stress resistance and phase variation of fimbriae to promote virulence.

### 3.5. Nutritional Stress and Metabolism

**The Pst system and the Pho regulon:** Phosphate, mostly inorganic phosphate (Pi), is implicated in several chemical reactions such as signal transduction by TCS. When phosphate is nutritionally limited (extracellular concentrations < 4 µM), Pi is transported by the Pst (for phosphate-specific transporter) system. The *pst* operon, *pstCAB-phoU* encodes an ABC transporter. This system has two clearly defined functions: (i) fixation of Pi and (ii) detection of Pi which regulates the expression of the Pho regulon. The Pst system is part of the Pho regulon, which is controlled by the TCS, PhoB/R [182]. Members of the Pho regulon are expressed under phosphate-deficient conditions but are repressed in a phosphate-enriched environment. In addition to its role in metabolism, Lamarche and colleagues have demonstrated that inactivation of the Pst system constitutively activates PhoB/R and attenuates virulence in a murine model [183]. In addition, inactivation of the *pst* system inhibited the expression of type 1 fimbriae in both APEC and UPEC strains [182,184]. Indeed, the *pst* mutant of UPEC strain CFT073 showed a decreased expression of the *fimA* structural gene which correlated with differential expression of genes encoding recombinases *fimB*, *fimE*, *ipuA*, and *ipbA* (Figure 1) [81].

**Frz:** In the APEC strain BEN2908, a genomic region involved in carbohydrate metabolism and transcribed in one operon called the *frz* operon was identified. The *frz* operon appears to be involved in the survival of the BEN2908 strain in LB medium during the late stationary growth phase under oxygen-restricted conditions. Results revealed that deletion of *frz* results in reduced production of type 1 fimbriae. Further, amplification of the *fimS* switch showed that the OFF orientation was increased in the mutant compared to the wild-type strain (Figure 1). In all, the *frz* operon plays a role in survival of ExPEC under stressful conditions such as oxygen and nutrient restriction and can contribute to virulence by promoting the expression of type 1 fimbriae [82].

One of the most important types of nutrient limitation in the host environment is metal limitation. Sequestration of metals by the host immune system has been termed “nutritional immunity”, since the availability of metals, such as iron and zinc, is required for microbial growth [155,185].

**Fur:** In bacteria, iron homeostasis is under the control of positive and negative regulators. Among these factors, the Fur (Ferric Uptake Regulator) transcription factor is highly conserved in many bacterial species. Fur represses genes encoding proteins involved in iron uptake and iron-dependent metabolic enzymes in an iron-rich environment. In UTIs caused by UPEC, a deletion of *fur* did not attenuate virulence. More specifically, the bacterial load of the *fur* mutant in the bladder was similar to that of the wild-type strain. Interestingly, the deletion of *fur* in the UPEC strain CFT073 results in increased adhesion and invasion of bladder epithelial cells in vitro [50], due to increased *fimA* expression and production of type 1 fimbriae (Figure 1). The absence of *fur* also led to increased IBC formation and the expression of *fliC*, a gene that contributes to bacterial motility. In fact, in iron-rich conditions, Fur protein is directly bound to the region upstream of *fimA* and *fliC*, resulting in a reduction of type 1 and flagellar gene expression. In summary, during UTI, bacteria are under restricted-iron conditions therefore, the Fur protein should remain inactive allowing expression of type 1 fimbriae, flagellar genes, and genes involved in biofilm formation [83].

### 3.6. Biofilm Formation

**Effect of salicylate on biofilm formation and relationship with MarA (UPEC):** MarA is an AraC/XylS transcriptional regulator that can repress or activate genes. For example, MarA is an activator of *tolC* and *ompX*. Salicylate represses the binding of MarR to the region of the *mar* operon. MarR proteins are members of the Multiple Antibiotic Resistance Regulator family of transcriptional regulators. This inhibition results in an increased production of MarA and a decreased accumulation of antibiotics that is associated with reduced production of OmpF and OmpC outer membrane porins and a concomitant increase in the production of the AcrAB multidrug efflux pump. In UPEC strain HC91255, RT-PCR and protein analyses showed that in the presence of salicylate, *marR* mutant strains overexpress *marA* and downregulate *fimA* and *fimB* expression and type 1 fimbriae production. The reduced production of recombinases leads to decreased production of type 1 fimbriae and biofilm formation [85].

**Biofilm and oxygen conditions:** UPEC are facultative anaerobic pathogens with great metabolic diversity. Previous studies have shown that deleting genes encoding enzymes in the tricarboxylic acid (TCA) cycle reduced virulence. The TCA cycle generates molecules such as NADH and FADH, which can be used in the electron transport chain if oxygen or one of the five alternative terminal electron acceptors (ATEA) is available. Further, UPEC mutants that are unable to use oxygen as a terminal electron acceptor seem to have attenuated virulence. These observations suggest that UPEC strain UTI89 uses aerobic respiration in the urinary tract.

As mentioned before, type 1 fimbriae are largely involved in biofilm formation, and indeed, reduced production of type 1 fimbriae also results in decreased biofilm formation on abiotic surfaces [186]. Indeed, in UPEC strain 536, regulation of type 1 fimbriae by LrhA has also been proven to influence biofilm formation [73]. Interestingly, Eberly et al. reported that *fim* expression is reduced in the total absence of O_2_ in UPEC UTI89 [84]. Mutants incapable of aerobic respiration also exhibit a defect in type 1 fimbriae production and biofilm formation, suggesting that anoxic conditions reduce the formation of biofilm due to decreased levels of type 1 fimbriae. Since *E. coli* uses ATEAs when O_2_ is not available, biofilm formation could have theoretically been restored under anaerobic conditions in the presence of these ATEAs; however, this was not the case. O_2_ was demonstrated to be the terminal electron acceptor that allows the most robust biofilm formation. Potentially, O_2_ improves biofilm production since bacteria find themselves in ideal conditions to proliferate and persist, suggesting that the bladder is an ideal environment for biofilm formation by UPEC strains [84].

**The link between the TCS QseC/B and quorum sensing in the formation of IBCs**: Common studies have established that the TCS, QseB/C, can respond to quorum sensing and is also involved in pathogenesis. Generally, QseC phosphorylates QseB (the response regulator) which leads to increased transcription of virulence genes [187]. In UPEC strain UTI89, analyses of single deletions of QseB/C demonstrated that a *qseC* mutant is reduced in the formation and maturation of IBCs, while a double deletion of *qseBC* or a single deletion of *qseB* has no impact on pathogenesis. Interestingly, in the absence of QseC, QseB remains constitutively active which leads to a downregulation of type 1 fimbriae expression. To further confirm the importance of QseC in type 1 fimbriae expression, PCR analyses of *fimS* demonstrated that in the *qseC* mutant, *fimS* was mostly oriented in the OFF phase. However, the interplay between QseB and QseC is more complex than a simple TCS. QseC has a dual role as a bifunctional sensor kinase/phosphatase by dephosphorylating QseB and restoring levels of type 1 fimbriae production. In summary, QseB/C is an example of a detection system of quorum sensing which is involved in IBC formation and expression of type 1 fimbriae [86].

### 3.7. Physical Cues and Regulation of Type 1 Fimbriae

So far, specific regulatory proteins affecting the expression of type 1 fimbriae have been described. However, fimbrial components and lectin-mediated adherence can also contribute to the production of type 1 fimbriae. Schwan and others have demonstrated that the binding of the FimH adhesin leads to a positive feedback loop by comparing *fim* gene expression in the presence of mannose-coated Sepharose beads and normal Sepharose beads. More specifically, when analyzing transcripts, in the presence of mannose-coated beads, levels of *fimB* transcripts increase and levels of *fimE* transcripts decrease, which explains the overall increase in type 1 fimbriae expression. Type 1 expression levels were decreased when the FimH binding pocket was mutated, suggesting that the binding of the *fimH* gene product can indirectly regulate phase-switching. Interestingly, in UPEC strains, the percentage of ON cells persisted even in an acidic environment (pH 5.5) whereas in the wild-type strain, the percentage dropped significantly [188]. In this case, it is suggested that virulence contributes to the ability of bacteria to adapt and withstand environmental stresses. Conversely, a lower pH environment has been described to decrease type 1 fimbriae expression [189]. Moreover, a study done by Tchesnokova and others has shown that the binding of FimH to an antibody raised against its lectin domain leads to an increase in mannose-specific binding, resulting in increased adhesion to the uroepithelium. This result further demonstrates that the regulatory activity of FimH binding may play a role in eliciting an immune response that enhances virulence [190]. In fact, studies have demonstrated that the FimH adhesin is an important inducer of the innate immune response, specifically activating natural killer cells, by binding to toll-like receptor 4 [191,192].

### 3.8. Shear Stress

Up to now, different chemical and physiological stresses have been discussed, but physical or mechanical stresses can also play a part in the regulation of adherence and type 1 fimbriae. Shear stress is caused by fluid flow which generates frictional forces. In UPEC, adherent bacteria need to withstand the flow of urine in order to remain attached and colonize the uroepithelium. Therefore, type 1 fimbriae and biofilm formation are important for resistance to mechanical stress. As mentioned in previous sections, fimbriated bacteria can form multi-layered colonies or biofilms as they multiply. Studies have demonstrated that biofilm formation provides significantly greater resistance to shear stress. The outer layer potentially acts as a protective barrier for underlying cells within the biofilm, allowing most bacteria to remain adherent, and their membranes remain intact. Mutants lacking type 1 fimbriae were not able to form biofilms and therefore, could not resist shear stress [193]. Interestingly, Thomas and others have demonstrated that mannose-coated surfaces have a stronger level of FimH-mediated bacterial binding/attachment as shear stress force increases, suggesting that FimH binding may be force-activated or force-enhanced [194]. Although it is unknown whether shear stress has a direct effect on fimbrial transcription or expression, this result still demonstrates that mechanical stress promoted by shear forces, which can be a physiological defense against bacterial infection, may actually promote FimH-mediated adherence and potentially favor increased expression of these fimbriae due to FimH lectin binding.

## 4. Link between Stress and Type 1 Fimbriae in Non-Pathogenic *E. coli*

Since type 1 fimbriae are also frequently produced by non-pathogenic *E. coli*, regulation of these fimbriae has often been investigated in *E. coli* K-12. For instance, studies showed different effects of temperature or growth medium on type 1 fimbriae. More specifically, growth in glucose inhibits type 1 fimbriae [195], while expression of *fimB* is regulated by *N*-acetylglucosamine (NagC regulator) and *N*-acetylneuraminic (NanR regulator) [196]. Growth in minimal medium at a high temperature (42 °C) increases FimB-mediated switching while FimE-mediated switching is favored in rich medium at low temperature (28 °C) [71]. Moreover, acetylphosphate (AcP) plays an important role during aerobic growth in excess of carbon and during mixed acid fermentation. Wolfe and colleagues demonstrated that AcP positively regulates genes required for assembly of type 1 fimbriae. Interestingly, the Fe-S metalloregulatory protein, IscR modulates gene expression in response to the iron-limiting environment as well as oxidative stress. In *E. coli* K-12 strain MG1655, analyses suggest that IscR, represses type 1 fimbriae expression by regulating *fimE* [197]. Further research done by McVicker and colleagues highlighted that the SlyA protein acts as an activator of type 1 fimbriae expression by inducing *fimB* expression [198]. Concerning inflammatory response, a release of sialic acid can be recognized by *E. coli* as an indicator of inflammation and the latter can suppress *fimB* expression through the binding of NanR affecting the phase variation of *fimS* from ON to OFF [199]. It is likely that such regulatory mechanisms may also apply to type 1 fimbriae of some pathogenic *E. coli* strains, although it is important to consider that regulation of type 1 fimbriae can involve numerous players and can vary considerably among different strains or pathotypes.

## 5. Conclusions

In summary, the regulation of type 1 fimbriae is complex, and involves not only a wide range of regulators, but also specific environmental cues. As there are many signals, bacteria need different systems to respond specifically to different cues or stresses. *E. coli* strains have developed intricate regulatory pathways in order to acquire specificity in their adaptation to environmental conditions and that can affect control of type 1 fimbriae expression which is ultimately mediated by the orientation of an invertible ON/OFF promoter-containing switch. Less optimal environments can impose stresses that can alter the levels of type 1 fimbriae production. By having many regulatory pathways associated with regulation of type 1 fimbriae, a combination of responses can lead to alterations in the levels of expression of type 1 fimbriae. Since most regulators cause changes in expression of multiple genes, broad-range regulation may then need to be more finely tuned through cross-regulation with other regulatory pathways. Such crosstalk between regulatory pathways can lead to tighter regulation of type 1 fimbriae, and adds another layer to the complexity of type 1 fimbriae regulation. Finally, during an infection, pathogenic strains of *E. coli* are likely to be in a less than optimal environment within the host tissues [200]. Therefore, stress signals present within the host environment and immune defense responses can influence bacterial gene regulation, and phase variation of fimbriae and adaptation to such stresses are an important mechanism linked to the virulence of pathogenic *E. coli*. Differential and coordinated expression of fimbrial adhesins provides bacteria with the ability to shift receptor target affinity and therefore, change tissue preference. Moreover, we must highlight the importance of the specific strain and the type of infection it is associated with. Depending on whether the strain in question is an APEC, UPEC, or NMEC, the regulation of type 1 fimbriae can differ and therefore, the effect of environmental stress on type 1 fimbriae regulation can also be distinct depending on the host species and type of infection. Moreover, considering the regulatory crosstalk within different systems (between type 1 fimbriae, P fimbriae, F1C fimbriae, flagella), the link of type 1 fimbriae/virulence and stress is even more difficult to identify, but nevertheless interesting to study.

Through this review, we sought to present information supporting the important connection between stress responses and the expression of type 1 fimbriae. Concrete examples were highlighted including direct regulators of both stress and type 1 fimbrial expression such as ppGpp, LrhA, RpoS, or cAMP. Interestingly, type 1 fimbriae appear to be an alert signal for the bacteria of certain environmental disorders as its expression is influenced by genes implicated in stress responses. In numerous cases, as highlighted above concerning *treA*, the TCS Cpx, or OmpA, the precise link between regulation of stress and type 1 fimbriae remains to be elucidated. At this point, the direct interactions leading to changes in type 1 regulation that are dependent on these systems remain to be discovered, but the attenuation of virulence observed in mutants could be explained by a complex and indirect regulation of type 1 fimbriae through a number of stress regulators, resulting in a decreased expression of type 1 fimbriae.

### Perspectives and Outstanding Questions

Further insight into the regulatory mechanisms controlling the expression of type 1 fimbriae could provide a means to identify cues to inhibit the expression of fimbriae and other virulence factors, leading to novel avenues to treat or prevent such infections. For now, knock-out or deletion mutants are commonly used to investigate gene function in bacteria. However, this method cannot be applied to genes that are essential for cell growth. An alternative approach would be to conditionally silence the gene, knocking down its expression, without altering the genome. Using knock-down mutants through antisense RNA methods would provide a more transient and refined model to study the regulatory pathway of type 1 fimbriae under different growth conditions. This method is performed by the expression of an antisense RNA which binds and leads to degradation of its target mRNA, leading to decreased gene product. Gene knock-down can be complete or partial. In *E. coli*, knock-down experiments have already been done [201,202,203] and this would be an interesting avenue of research to pursue. Thus far, the phenomenon of phase variation of type 1 fimbriae has mainly been investigated in the context of a global population of multiple bacterial cells. With newer approaches and technologies now available to investigate single-cell gene expression and adaptation, it will be of interest to determine mechanisms of regulation and regulation of the switching of the *fim* promoter switch at the single-cell level as well as within a population of *E. coli* cells.

## 6. Glossary

**ABC transporter (ATP-Binding Cassette Transporter):** A member of a ubiquitous superfamily of membrane-bound pumps present in all prokaryotes. Directional substrate transport across a membrane bilayer is achieved by an ATP-dependent flipping mecha-nism from an inward- to an outward-facing conformation.

**Adhesin:** The surface-exposed bacterial molecule that mediates specific binding to a receptor or ligand on a target cell.

**Biofilm:** A community of cells that are attached to a surface or interface or to each other, and are imbedded in a self-made, protective matrix of extracellular polymeric sub-stances that are protected from immune responses, antimicrobial agents, and other stresses.

**CCR (Carbon Catabolite Repression):** A regulatory phenomenon by which the ex-pression of functions for the use of secondary carbon sources and the activities of the cor-responding enzymes are reduced in the presence of a preferred carbon source.

**CUP (Chaperone Usher Pathway):** A system that facilitates the folding, transport and ordered assembly of fimbriae subunits at the cell surface.

**Fimbriae:** Long non-flagellar appendages at the cell surface, also referred to as pili, that are present in a wide range of Gram-negative and Gram-positive bacteria and in ar-chaea, and are involved in bacterial attachment.

**Curli:** Extracellular amyloid-like protein fibres produced by some bacteria, which are involved in adhesion, biofilm formation, and surface colonization.

**HSR (Heat Shock Response):** A response which involves the induction of expression of a large number of proteins upon increases in temperature.

**IBC (Intracellular Bacterial Communities):** Uropathogenic *Escherichia coli* (UPEC) biofilm-like intracellular bacterial communities formed by uropathogenic *Escherichia coli* that protect their members from the immune system, antibiotics, and other stresses.

**Iron requirement:** One of the most important types of nutrient limitation in the host environment is metal limitation.

**Nutritional immunity:** Sequestration of metals by the host immune system, leading to nutrient-limited environments that antagonize bacterial survival.

**SecYEG translocon:** A conserved machinery that mediates the translocation of pro-teins across biological membranes and into different cellular compartments.

**Shear stress:** Forces that are applied tangentially to a body’s surface, generally gen-erated by flow.

## Figures and Tables

**Figure 1 microorganisms-10-00005-f001:**
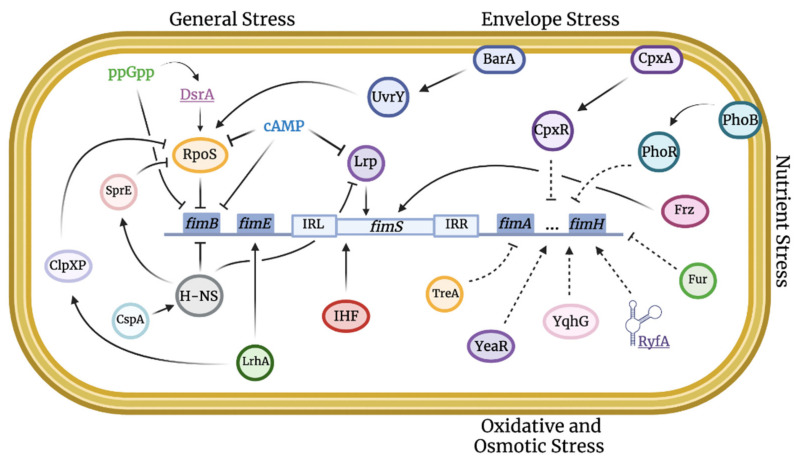
**Integration map of stress-induced pathways implicated in type 1 fimbriae regulation.** Stress regulation can be linked to virulence, such as the expression of type 1 fimbriae, through an intrinsic network of direct and indirect pathways. Solid lines indicate confirmed stimulatory or inhibitory effects. Dashed lines indicate unclear mechanisms that remain to be elucidated.

**Figure 2 microorganisms-10-00005-f002:**
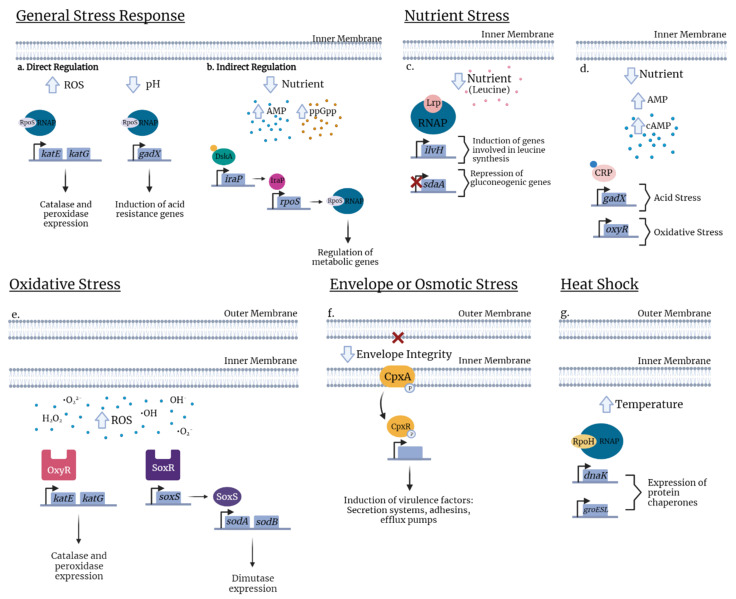
Examples of stress regulators in *E. coli*. **General stress response.** (**a**) In response to oxidative stress, RpoS occurs in direct regulation by binding to RNA polymerase (RNAP) and recognizes the promoter thus allowing expression of *katG* and *katE* catalase and peroxidase expression. Likewise, in response to low pH, binding of RpoS to RNAP induces expression of the transcriptional regulator, *gadX*. (**b**) Under nutrient limitation, RpoS is indirectly regulated by the transcription factor DskA or by the alarmone ppGpp (orange circle) that leads to the augmentation of the anti-adaptor IraP and releases RpoS to activate stress gene expression. **Nutrient stress.** (**c**) Under nutrient deficient conditions, a mis-regulation of cAMP signaling for nutrient availability allows binding of cAMP to the cAMP Receptor Protein (CRP) which activates the protein and specific binding with target DNA sequences regulating the expression of genes involved in acid stress (*gadX*) or in oxidative stress (*oxyR*). (**d**) In nutrient deprivation, exogenous leucine (pink circle) influences the Lrp regulon and modulates Lrp directly. Presence of leucine concentrations represses the transcription of the *ilvH* promoter whereas in the absence of leucine, *ilvH* is directly activated by Lrp. Inversely, leucine releases Lrp to bind to the *sdaA* promoter and activates its expression. **Oxidative stress.** (**e**) In response to oxidative stress due to excess levels of prooxidants (H_2_O_2_, O_2_, OH), depending on whether the stress is mediated, bacteria respond by two regulatory systems, the peroxide regulon (OxyR) or the superoxide regulon (SoxR/S). OxyR activates genes involved in catalase and peroxidase expression (*katE* and *katG*). When oxidized, the sensor SoxR activates *soxS* transcription resulting in expression of superoxide dismutase (*sodA* and *sodB*). **Envelope stress.** (**f**) The two-component system consists of the inner membrane, the sensor histidine kinase (CpxA) and the cytoplasmic response regulator CpxR. Envelope stress conditions lead to phosphorylation of CpxA which transfers the phosphate group to CpxR. Phosphorylated CpxR-P functions as a transcriptional regulator which controls the expression of numerous genes including some virulence factors. **Heat shock.** (**g**) In a simple pathway, during temperature upshift (30 °C to 42 °C), the Heat Shock Response (HSR) is induced by the increase of RpoH levels, primarily due to an enhanced translation of rpoH mRNA and stabilization of the protein. The elevated temperature disturbs protein homeostasis and induces accumulation of misfolded proteins. Chaperones DnaK and GroEL/S which are proteins helping to activate or degrade RpoH and regulate heat shock gene transcription.

**Figure 3 microorganisms-10-00005-f003:**
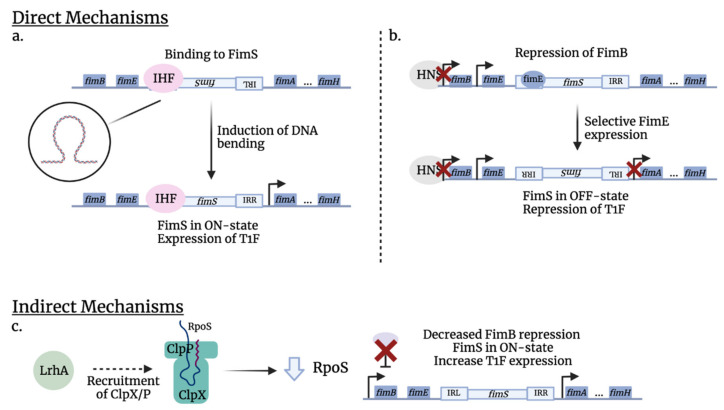
Mechanisms of action of stress regulators on type 1 fimbriae expression. (**a**) Regulators can bind directly to *fimS* to influence phase variation. For example, IHF binds to regions of *fimS* to induce a sharp DNA bend that facilitates the recombination of the *fim* switch. In nutrient-deprived environments, such as when bacteria enter the stationary phase, IHF expression will be induced, increasing phase variation, and influencing type 1 fimbriae expression. (**b**) Regulators can block recombinase expression to influence *fimS* phase variation. In the case of H-NS, the regulator can bind to the FimB promoter to block its expression, resulting in more FimE production. Since FimE facilitates phase variation of *fimS* in the OFF-state, type 1 fimbriae will be repressed. This is a simplified model of regulation by H-NS as the latter binds to both FimB and FimE promoters. (**c**) Other regulators may be indirectly linked to stress by influencing direct stress regulators. Although LrhA can act directly on type 1 fimbriae and flagellar gene expression, it is indirectly linked to stress via the RpoS network. LrhA recruits ClpX/P protease complex through an unknown mechanism (dotted arrow), leading to reduced RpoS production. As a result, FimB repression decreases and more *fimS* is found in the ON-state, which increases type 1 fimbriae expression.

**Table 1 microorganisms-10-00005-t001:** Global and specific stress response regulators involved in virulence and virulence gene expression in *Escherichia coli*.

Regulator	Stress Response	Role in Virulence	Reference
RpoS	Nutrient deprivation	Master regulator of stress	[41]
H-NS	Temperature	Regulates flagellar gene expression and *fim* and *pap* operon and many other genes	[42]
Lrp	Nutrient deprivation	Required for *fim* and *pap* fimbriae	[43]
ppGpp	Stringent response	Involved in biofilm formation and production of flagella	[44,45]
cAMP	Nutrient deprivation	Required for acid stress response, regulation of multiple virulence factors	[46]
SoxS/R and OxyR	Oxidative stress	Required for virulence in UPEC	[47]
CpxRA	Membrane damage	Required for type 1 and P fimbriae expression in UPEC	[48]
sRNA	Diverse	MicF regulates gene expression for the outer membrane	[49]
		RyhB is required for nutrient stress/iron homeostasis	[50]
		GadY is required for acid stress resistance	[51]
		RyfA is required for survival in human macrophages, resistance to multiple stresses	[52]
RpoH	Heat shock	Regulates gene expression in heat shock	[53]

**Table 2 microorganisms-10-00005-t002:** Example of regulators of type 1 fimbriae in ExPEC involved in stress resistance.

Regulator	Switch	FimE	FimB	Effect on *Fim* Expression	Reference
**General and specific stress regulators**					
IHF	Switching on *fimS*			Positive or negative ^1^	[70]
Lrp		+/−	+/−	Positive or negative ^1^	[71]
H-NS		−	<37 °C: − >37 °C: +	<37 °C: Negative >37 °C: Positive	[72]
RpoS			−	Negative	[66]
LrhA		+		Negative	[73]
ppGpp			−	Negative	[67]
cAMP			−	Negative	[68]
**Envelope stress**					
CpxR-P	Regulates the inversion		-	Negative	[74]
BarA/UvrY	Reduction of *fimA*	Unknown		Unknown ^1^	[75]
**Oxidative and osmotic stress**					
TreA		Unknown	Unknown	Positive	[76]
YeaR		Unknown	Unknown	Positive? ^1^	[77]
IbeA		+ ?	+ ?	Positive? ^1^	[78]
YqhG		Unknown	Unknown	Positive? ^1^	[79]
RyfA		Unknown	Unknown	Positive? ^1^	[52]
**Nitrosative stress**					
FimX		Unknown	Unknown	Positive?	[80]
**Nutrient limitation and oxygenation**					
Pst and Pho regulon		+	−	Negative	[81]
Frz		Unknown	Unknown	Positive	[82]
Fur	Increased *fimA*		Unknown	Positive	[83]
Oxygenation		Unknown	Unknown	Positive	[84]
**Biofilm and quorum sensing**					
Effect of salicylate on *marA*			−	Negative	[85]
QseC/B		Unknown	Unknown	Positive	[86]

^1^ Putative role.

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
