# Peer review of "What Flips the Switch? Signals and Stress Regulating Extraintestinal Pathogenic Escherichia coli Type 1 Fimbriae (Pili)"

_microorganisms, 2021, doi:10.3390/microorganisms10010005_

Round 1
Reviewer 1 Report
In this review, titled “What flips the switch? Signals and stress regulating E. coli type 1 fimbriae (pili)”, Bessaiah et al, aim to summarize what is known about type 1 fimbriae in ExPEC. This work is useful for the scientific community but many issues need to be addressed before being accepted for publication.
The article title is about E. coli in general, however, the authors discussed mainly ExPEC (UPEC and NMEC). As stated in the article, ExPEC are phylogenetically and epidemiologically distinct from other E. coli, meaning that what is true for ExPEC is maybe not true for other E. coli. So the article name and the abstract should be modified or more references and discussion should be added about the other E. coli like EHEC, EPEC, ETEC…
The authors claim that ExPEC expresses a large number of fimbrial adhesins (Page 9). Do they have any references showing other (than Type 1 fimbriae) fimbrial expression? This ref should be cited: 10.1111/j.1758-2229.2010.00166.x
The authors should be cautious with this claim, as little is known about many fimbriae (at protein level) expressions under different growth conditions.
Chapter 4 is not about the role of type 1 fimbriae from ExPEC but more about the type 1 fimbriae synthesis. This chapter should be renamed and a new one added to discuss the actual role in the E. coli pathogenicity of these fimbriae (Mannose recognition, where are the glycans recognized in the host…).
Page 4, CUF are described as the most common fimbrial type encoded by E. coli. The authors cite a paper from 1965. In the light of the identification of many other adhesins (Curly, TAA…), I am not sure about the accuracy of this claim.
Few words about pili and fimbriae could help the reader to understand why "pili" is in brackets.
This ref: 10.1093/femsle/fnw251, could be added in chapter 12.
It could be interesting to add in the conclusion some hypotheses about the need for so many regulatory pathways associated with type 1 fimbriae.
Please revise the format of the references by removing the not-needed uppercases (ref 10, 14, 18…).
Author Response
Thank you for your review and supportive critiques of our manuscript:
Please find a point-by-point response to your comments.
...The article name and the abstract should be modified or more references and discussion should be added about the other E. coli like EHEC, EPEC, ETEC…
- We thank the reviewers for their excellent reviews and agree that the title would be more specific for ExPEC. The new title is<« What flips the switch? Signals and stress regulating Extraintestinal pathogenic Escherichia coli type 1 fimbriae (pili)»
The authors claim that ExPEC expresses a large number of fimbrial adhesins (Page 9). Do they have any references showing other (than Type 1 fimbriae) fimbrial expression? This ref should be cited: 10.1111/j.1758-2229.2010.00166.
- We have added this information and references Paragraph 2, page 2
Page 4, CUF are described as the most common fimbrial type encoded by E. coli. The authors cite a paper from 1965. In the light of the identification of many other adhesins (Curly, TAA…), I am not sure about the accuracy of this claim.
- This was an error and we have changed this.
Few words about pili and fimbriae could help the reader to understand why "pili" is in brackets.
- We have explained both word pili and fimbriae (end of Paragraph 2, page 2)
It could be interesting to add in the conclusion some hypotheses about the need for so many regulatory pathways associated with type 1 fimbriae.
- Thank you for the suggestions. We edited the conclusion and added some new ideas. We added a last part about perspectives & outstanding questions
Reviewer 2 Report
In this review authors summarized different stress responses and the complex regulation of type 1 fimbriae expression in pathogenic Escherichia coli. Overall, the review is well written and very informative. However, the clarity is lost in some areas due to too much information. To improve the clarity and succinctness, authors need to focus on the topic – regulation of type 1 fimbriae expression. Below are few suggestions for authors –
- Need to reduce the introduction of bacterial different stress responses, need minimal information.
- Detail introduction of type 1 fimbriae and its importance in pathogenesis.
- Need to bring Figure 3 on top and summarize the overall mechanism of regulation of type 1 fimbriae expression.
- Then detail discussion on individual mechanisms.
- All sections need a short title.
- All distantly related topic and terminology need to add as separate glossary or box, like the style of “Nature Reviews of Microbiology”.
- Finally, need to summarize all the key points under conclusion.
- Need to also include the outstanding questions.
Author Response
We thank the reviewer for their comments and expert evaluation of our manuscript. Overall, we have made substantial changes and have responded to specifically address all the comments and critiques that were raised by the reviewers below.
Here is a point-by-point response to the specific comments.
- Need to reduce the introduction of bacterial different stress responses, need minimal information.
- To address these comments, we have rewritten and re-organized some sections of the manuscript, including introduction, have included additional information and revised the discussion as requested.
2. Detail introduction of type 1 fimbriae and its importance in pathogenesis.
- We have added a section about Role of type 1 fimbriae in pathogenesis (page 2 paragraph 3)
3. Need to bring Figure 3 on top and summarize the overall mechanism of regulation of type 1 fimbriae expression. 4. Then detail discussion on individual mechanisms.
- We have changed the order of the figures and summarize the overall mechanism of regulation of type 1 fimbriae , followed by the detai of individual regulatory mechanisms
5. All sections need a short title.
- We have added short titles
6. All distantly related topic and terminology need to add as separate glossary or box, like the style of “Nature Reviews of Microbiology”.
- Thank you for the suggestion. We added a glossary in the end of the manuscript
7. Finally, need to summarize all the key points under conclusion.
- As requested by the reviewer, we have now included key points in the conclusion
8. Need to also include the outstanding questions.
- We have added the requested part about perspectives & outstanding questions at the end of the manuscript
Round 2
Reviewer 1 Report
The authors have modified the paper as requested.
Reviewer 2 Report
The revised version of the manuscript appears to be improved.